# BASE MODELS KNOW HOW TO REASON, THINKING MODELS LEARN WHEN

## ABSTRACT

Why do "thinking" language models like DeepSeek R1 outperform their base counterparts? Despite consistent performance gains, the underlying mechanism, i.e., what differs between base and thinking models, remains unclear. In this work, we show that, by deriving mechanism-specific steering vectors and activating the right vector at the right time in base models, we can elicit thinking-model-level reasoning chains. To ground our analysis, we introduce an unsupervised, bottom-up approach for uncovering human-interpretable reasoning mechanisms in thinking models. This approach provides a principled way to discover reasoning mechanisms without imposing manual or LLM-derived assumptions about what reasoning mechanisms should exist, reducing bias. Across three base and four thinking models, on GSM8K and MATH500, our combined approach substantially lifts base-model accuracy, recovering up to $91\%$ of the gap to thinking models without any weight updates. This suggests that reinforcement learning with verifiable rewards mainly teaches thinking models when to fire pre-trained skills, rather than how to execute them, enabling the model to productively use its inference-time compute. Our work reframes our understanding of the performance gains of thinking models and may shed light on why distillation and reinforcement learning are so effective for teaching reasoning to small LLMs.

## 1 INTRODUCTION

Large Language Models (LLMs) have recently demonstrated remarkable capabilities in reasoning tasks when given additional inference time to think through problems step-by-step. *Thinking models*, also known as *reasoning models*, or *models using inference-time compute*, are a type of language model designed to generate long chains of reasoning before arriving at a final answer. Examples of models in this category include Anthropic's Claude 3.7 Sonnet (Anthropic, 2025), OpenAI's o3 (OpenAI, 2025), Gemini 2.5 Pro (Google, 2025), DeepSeek's R1 (DeepSeek-AI, 2025), and Qwen's QwQ 32B (Qwen Team, 2024).

All these *thinking* models significantly outperform their base counterparts on challenging reasoning benchmarks (Chollet, 2024). However, a fundamental question remains: *What is the difference between base and thinking models that allows the latter to achieve superior performance?*

Prior work has suggested several hypotheses: (1) *thinking* models acquire entirely new reasoning capabilities through specialized training (Gandhi et al., 2025); (2) reinforcement learning (RL) teaches them to structure their reasoning more effectively (Marjanović et al., 2025); (3) RL teaches them to repurpose pre-existing base model representations for new mechanisms (Ward et al., 2025); or (4) the additional inference time simply allows more computation to be applied to difficult problems (Zhao et al., 2025; Wang et al., 2025). In this paper, we present evidence for a more nuanced explanation: *not only do base models already possess the fundamental reasoning capabilities, but thinking models learn when to deploy these capabilities in a structured sequence.*

We make the following contributions:

1. To support our analysis, we develop an unsupervised clustering methodology to derive an interpretable taxonomy of reasoning mechanisms that thinking models employ during their chains of thought, forming the building blocks of complex problem-solving (Section 2).

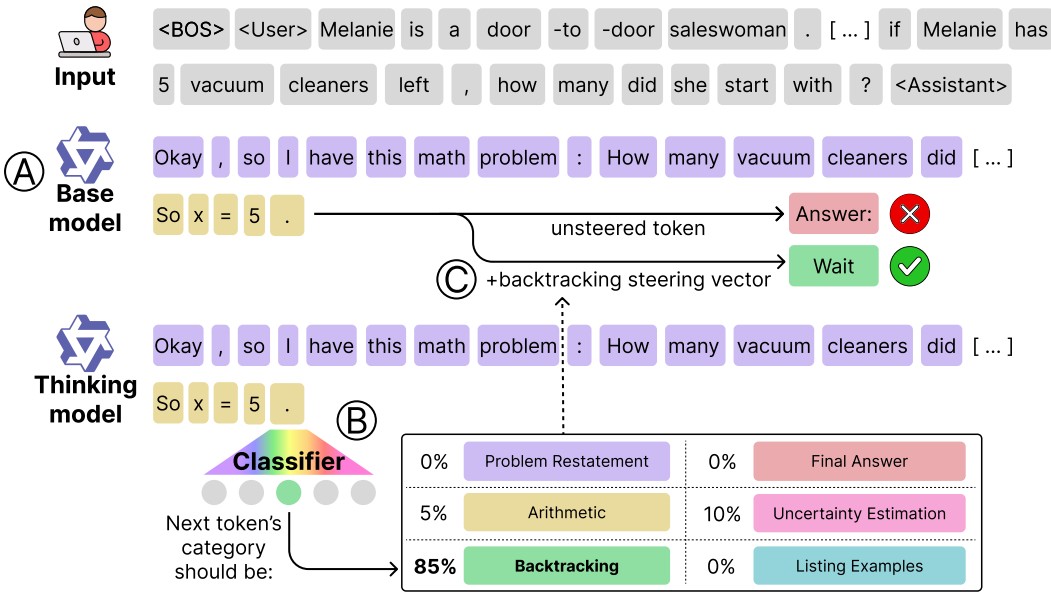

Figure 1: **Hybrid Models Unlock Reasoning Model-Level Behavior with Minimal Intervention.** Overview of our approach for steering base language models to reason like *thinking* models. **(A)** We use the base model as the primary generator of tokens in the output sequence. **(B)** For each token position, we evaluate the current rollout in a target thinking model and use a "thinking model activation classifier" to detect the reasoning mechanism that should be applied next. **(C)** When the classifier detects a reasoning step, we apply a corresponding steering vector to the base model's activations, triggering structured reasoning behavior. This approach shows that base models already possess latent reasoning abilities, and that these can be reliably activated without any parameter updates, bridging much of the gap to full reasoning models with minimal extra machinery.

2. We demonstrate that **base models can perform each reasoning behavior when appropriate steering vectors are added to their activations** (Section 3). By identifying and applying the right vector at the right step, we can guide pretrained base models to reproduce complete reasoning chains of thinking models.

The evaluation for our steered base model approach focuses on thinking models across diverse architectures and parameter scales, including models trained with distillation (DeepSeek-R1-Distill series) and models trained directly with RLVR (QwQ-32B). The results show that this approach substantially lifts base model performance, recovering up to $91\%$ of the performance gap between base and thinking models on GSM8K and MATH500 benchmarks *without any weight updates*.

This finding provides strong evidence that reinforcement learning with verifiable rewards (RLVR (Yue et al., 2025)) used to train thinking models primarily teaches *when* to activate pre-existing skills rather than teaching *how* to execute those skills. This perspective has direct implications for more efficient training of reasoning in future language models. To ease reproducibility and further research, we publish our codebase and results in a public GitHub repository[1].

## 2 TAXONOMY OF REASONING MECHANISMS

Recent work on thinking models has primarily relied on manual inspection of the model's reasoning traces to identify the underlying mechanisms it uses to perform reasoning (see Section 4). While insightful, such approaches are inherently subjective and may overlook subtle or distributed reasoning patterns. To support our main analysis, we develop an unsupervised, bottom-up methodology to

---

[1][Withheld for anonymous review]

discover human-interpretable reasoning mechanisms in thinking models. Our goal is to construct a taxonomy of reasoning mechanisms that is:

1. **Interpretable:** Each reasoning mechanism should be understandable by humans, with a clear description of its cognitive function and role in the reasoning process.
2. **Complete:** The taxonomy should cover the full range of types of reasoning steps the model can use, ensuring no significant patterns of reasoning are overlooked in our analysis.
3. **Independent:** The categories should correspond to distinct cognitive functions with minimal overlap between different reasoning processes.

### 2.1 Unsupervised Clustering of Reasoning Mechanisms via High-Level Sparse Autoencoders

Unsupervised methods are essential for building a taxonomy of reasoning mechanisms. They allow us to discover reasoning patterns without imposing pre-existing assumptions about how models reason, which could bias our taxonomy. Clustering algorithms are particularly well-suited as they can identify natural groupings in high-dimensional activation spaces that correspond to distinct reasoning functions.

Sparse Autoencoders (SAEs) (Olshausen & Field, 1997; Lee et al., 2006) have gained widespread popularity in recent years due to their ability to decompose Large Language Model (LLM) activations into interpretable features (Cunningham et al., 2023; Bricken et al., 2023; Templeton et al., 2024). Top-K SAEs (Makhzani & Frey, 2014; Gao et al., 2024) are a variant that enforces sparsity by keeping only the $K$ largest magnitude components of the latent representation, creating a more interpretable and computationally efficient decomposition. In our approach, we use Top-K Sparse SAEs to cluster the sentence-level activations of the model. More details on how Top-K SAEs operate are provided in Appendix A.

The configuration of our SAE directly matches our hypotheses about reasoning processes. Specifically, the dimension size of the SAE dictionary represents the number of distinct reasoning mechanisms we hypothesize exist in the model's reasoning process, while the parameter $k$ in top-$k$ sparsity constrains how many reasoning mechanisms can be simultaneously active in a single sentence. This reflects our hypothesis that each reasoning step typically employs a small number of distinct cognitive operations rather than engaging all possible reasoning mechanisms at once.

Using a restricted decoder space, we force the SAE to learn the subspace components that best explain the variance of our sentence activations, essentially making this a clustering method. While standard configurations in Mechanistic Interpretability typically use much larger latent dimensions than input dimensions (Templeton et al., 2024; Gao et al., 2024), we deliberately restrict the latent dimension to be in the range $[5, 50]$, which is far smaller than the input dimension (e.g., 1,536 for Qwen2.5-1.5B). This design choice forces the SAE to identify the most fundamental dimensions of reasoning variation rather than incidental linguistic features. This constraint ensures discovered features correspond to core cognitive operations, with empirical evidence showing that optimal dimensionality consistently falls between 15-25 categories across different model architectures.

We train our Top-K Sparse Autoencoders (SAEs) on sentence-level activations extracted from reasoning traces generated on 12,102 prompts from MMLU-Pro (Wang et al., 2024), resulting in 430,122 sentences. We focus on sentence-level analysis because sentences strike an intermediate abstraction depth that is optimal for reasoning analysis, avoiding the excessive granularity of token-level analysis while maintaining more precision than paragraph-level approaches (Bogdan et al., 2025). Prior work has established that different sentences within reasoning traces perform distinct functions (Bogdan et al., 2025; Venhoff et al., 2025), providing computational tractability for attribution and causal analysis across long traces (Nye et al., 2021). We average activations over sentences under the assumption that each sentence can be primarily classified by one or, at most, three reasoning categories. More details on the training process of SAEs are provided in Appendix A.

### 2.2 Taxonomy Evaluation

Once we obtain SAE clusters for each model configuration, we need to systematically evaluate the quality of the resulting taxonomy against our stated objectives of interpretability, completeness, and independence. Given that we do not know *a priori* how many clusters are optimal for each model or

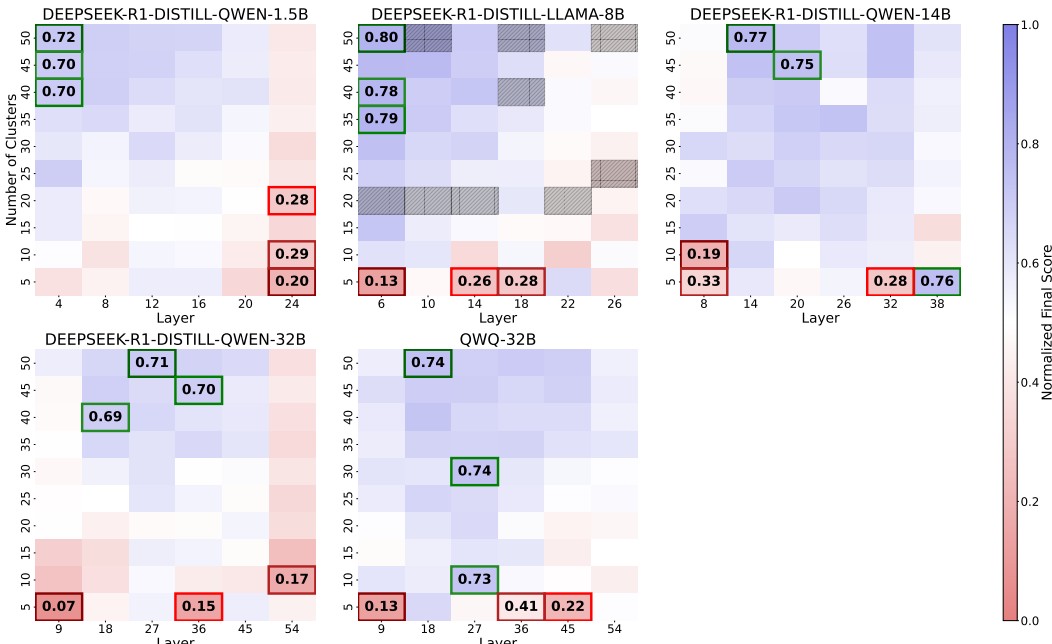

Figure 2: **Grid search results** comparing the performance of Sparse Autoencoder taxonomies across the five thinking models in our taxonomy evaluation. Each heatmap shows the combined score (average of completeness, independence, and consistency) for different combinations of layer locations (x-axis) and dictionary sizes ranging from 5 to 50 with increments of 5 (y-axis). Values highlighted in blue indicate particularly strong performing configurations, while red ones indicate poor-performing configurations. Optimal configurations are typically found in the upper portions of the grid (0.70-0.80). Despite the common pattern of high scores on large cluster sizes, we find "elbow" scores at cluster sizes between 10 and 20, suggesting that **reasoning mechanisms are reasonably well represented using 10 to 20 categories**.

which layer would best capture the reasoning mechanisms, we need a robust methodology to evaluate and compare different SAE configurations. We implement this evaluation as a scoring system that allows us to identify the most effective taxonomy across different hyperparameter settings. For this evaluation, we used the sentences we collected from the reasoning traces of each thinking model on MMLU-Pro prompts. Each potential taxonomy in our work is evaluated through a scoring metric defined as an average of three components: *completeness*, *consistency*, and *independence*.

**Interpretable Categories.**  To derive human-understandable categories from our Sparse Autoencoder (SAE) representations, we employed an LLM-based interpretability approach. Our SAE-based approach offers a key advantage over direct LLM annotation of reasoning patterns: it provides fully unsupervised discovery of categories without imposing manual or LLM-derived assumptions about what reasoning mechanisms should exist, making it more principled and unbiased. For each identified cluster, we collected 100 top exemplar sentences that most strongly activated that particular feature, and 100 random sentences from the same cluster. These representative examples were then analyzed using an LLM to identify the precise cognitive function these sentences serve in the reasoning process.

This process generates our list of interpretable reasoning categories with their titles and descriptions, which forms the foundation for our subsequent evaluation metrics. See Appendix B.1 for the complete cluster generation prompt and more details.

**Consistency.**  We measure the consistency of our categories by evaluating how well an LLM can classify individual sentences from within and outside each category using the generated titles and descriptions. For a given cluster, we take the average F1 score across all categories as our overall consistency score.

**Completeness.**  We measure the completeness of our categories by evaluating the confidence that an LLM has in classifying individual sentences into their assigned categories.

**Independence.**  We measure the independence of our categories by asking an LLM to evaluate how semantically similar all pairs of categories are in a cluster. This information is then used to calculate the fraction of pairs with similarity below a threshold ($0.5$), equivalently, those with orthogonality above $0.5$, which we consider functionally distinct.

For all these metrics, the higher the value, the better the taxonomy. More details on the prompts and specific implementation are provided in Appendix B. Note that the consistency, completeness, and independence scores are generated by prompting an LLM. Although LLM-as-a-judge is a common practice in current literature, the alignment between our evaluation pipeline and true human judgment remains to be validated.

As mentioned, the final score for a cluster is calculated as the average of the three evaluation metrics, providing a systematic way to evaluate the quality of our taxonomy, and ensuring it effectively captures the full spectrum of reasoning mechanisms while maintaining clear boundaries between categories.

Additionally, this approach enables us to move beyond manual, top-down annotation and instead discover reasoning categories that are both grounded in the model's internal representations and interpretable to humans. We note that other clustering approaches or manually designed sentence taxonomies would likely work similarly for our purposes, as our main findings depend on the existence of reasoning categories rather than their specific derivation method.

### 2.3 Taxonomy Results

To evaluate our approach to building interpretable taxonomies, we analyze five models: four DeepSeek-R1 distilled variants: `Llama-8B`, `Qwen-1.5B`, `Qwen-14B`, `Qwen-32B`, and one model trained with Reinforcement Learning from Verifier Rewards (RLVR), `QwQ-32B`.

The DeepSeek-R1 distilled models are a series of smaller dense models, which have been fine-tuned to mimic the behavior of the full DeepSeek R1 model, a recent *thinking* model that has achieved a similar performance to OpenAI's o1-preview on the ARC-AGI-Pub dataset (Knoop, 2025; Chollet, 2024). These distilled models have parameter counts ranging from 1.5B to 70B, implemented on both Qwen and Llama architectures, and their performance matches or exceeds that of leading production models, including GPT-4o (OpenAI, 2024) and Claude 3.5 Sonnet (Anthropic, 2024), across several math and coding benchmarks (DeepSeek-AI, 2025).

The DeepSeek-R1 model itself is trained using a multi-stage process that combines large-scale reinforcement learning (RL) with supervised fine-tuning (SFT). Similarly, `QwQ-32B` is a large language model trained with RLVR, which optimizes the model with stepwise signals from automated verifiers rather than outcome-only rewards, explicitly shaping intermediate reasoning. This model is not distilled from DeepSeek-R1, providing a contrast between distillation-driven and verifier-driven reasoning.

We performed an extensive grid search across these five models, using 6 distributed layers and dictionary sizes (ranging from $5$ to $50$ categories with increments of $5$) to identify the optimal taxonomy configuration. For comparison across configurations, we apply min-max normalization within each model. The results are shown in Figure 2, and we provide the complete taxonomies for our best-performing SAE configurations in Appendix E.

## 3 Steering Base Models to Reason

In this section, we explore the main question of our paper: *do base models already possess the reasoning mechanisms of thinking models, and if so, can we induce these behaviors through targeted interventions?* Our hypothesis, supported by preliminary evidence in prior work (Ward et al., 2025; Hou et al., 2023; Galichin et al., 2025), is that non-thinking models may already contain the latent capacity for sophisticated reasoning patterns, such as uncertainty estimation and backtracking, but lack the ability to effectively determine when to employ these mechanisms.

Following Marjanović et al. (2025), we define a *reasoning behavior* (or *reasoning mechanism*) as an individual cognitive-like step or operation that a model performs as part of its chain-of-thought when working through a problem. Such steps, for example, verifying an intermediate result, backtracking to revise an approach, or setting a subgoal, serve as interpretable, compositional building blocks of the model's reasoning process.

To investigate this hypothesis, we propose a **hybrid approach** that combines the strengths of base models with the decision-making capabilities of thinking models. We control the base model with steering vectors: directions in activation space that, when added to intermediate activations, induce target behaviors (Turner et al., 2023; Arditi et al., 2024; Zou et al., 2023; Panickssery et al., 2023). This leverages the linear representation hypothesis, which posits that certain concepts and behaviors in neural networks are represented as directions in activation space. The details of how we find and compute the steering vectors are provided in Appendix C.

Once we have extracted the causal vectors that induce the reasoning mechanisms in base models using the approach in Section 2, we allow a thinking model to decide when to activate these steering vectors by analyzing the base model's generation and identifying appropriate moments to induce specific reasoning mechanisms. This flow is depicted in Figure 1.

If this hybrid model performs comparably to dedicated thinking models, it would provide evidence that the fundamental reasoning mechanisms already exist within base models, and that thinking models primarily learn when to optimally deploy these mechanisms rather than developing entirely new capabilities.

## 3.1 FINDING STEERING VECTORS IN BASE MODELS

We leverage the reasoning taxonomies we built in Section 2 to identify steering vectors corresponding to each reasoning mechanism. For a given reasoning category, the steering vector represents the direction in activation space that induces the corresponding behavior in the base model. Since SAEs identify variance-explaining rather than causally important directions, we use steering vector optimization to search for the causal directions corresponding to SAE-discovered reasoning mechanisms in base models.

To find steering vectors through optimization, we employ the method outlined by Dunefsky & Cohan (2025). Specifically, we:

1. Choose an SAE from Section 2 to label each sentence in our dataset of reasoning traces from MMLU Pro tasks (Wang et al., 2024), with its corresponding reasoning category.
2. For each category, identify sentences with top activation scores for that category.
3. Extract examples where we have both the prefix leading up to the annotated sentence and the annotated sentence itself as the target completion.
4. Optimize a steering vector in the base model that, when applied, maximizes the next token prediction loss for the thinking model's completion while minimizing the likelihood of the base model's completion.

Based on our grid search results shown in Figure 2, we select for each model a layer and cluster size that lies at the performance elbow, providing a practical balance between completeness and independence while avoiding the computational overhead of larger cluster sizes. We optimize steering vectors at 37% of model depth, which has been shown to be most causal for some models in prior work (Venhoff et al., 2025) (e.g., for Llama-3.1-8B distilled, layer 12). Notice that this depth is different than the layer used for evaluating SAE layers in the taxonomy extraction. The complete procedure for example selection and training is detailed in Appendix C.

In addition to category-specific steering vectors, we train a general *bias vector* using a randomly sampled set of thinking rollouts as the target completion. This bias vector is supposed to capture general similarities across complete rollouts, like using first person. We apply it during the generation of the hybrid model, frozen, alongside category-specific vectors during training. During training, we apply the steering vectors at all token positions. The complete training procedure, including optimization hyperparameters, early stopping criteria, and prompt templates, is detailed in Appendix C. The steering vectors converge successfully during training across different model architectures and sizes, indicating that base models contain causal directions that can reliably steer them toward the reasoning behaviors discovered by the SAE.

Table 1: **Hybrid model performance on GSM8K.** Results show accuracy percentages for base models, hybrid models (base + steering vectors), and thinking models. Performance improvements over base model are shown in parentheses next to hybrid and thinking model results.

| Base Model | Thinking Model | Base | Hybrid | Thinking | Gap Recovery |
|---|---|---|---|---|---|
| Llama-3.1-8B | DeepSeek-R1-Distill-Llama-8B | 31.7% | 63.3% (+31.6%) | 80.8% (+49.1%) | **64**% |
| Qwen2.5-14B | DeepSeek-R1-Distill-Qwen-14B | 90.3% | 92.2% (+1.9%) | 95.0% (+4.7%) | **40.0**% |
| Qwen2.5-32B | DeepSeek-R1-Distill-Qwen-32B | 93.0% | 96.7% (+3.7%) | 94.4% (+1.4%) | **266.7**% |
| Qwen2.5-32B | QwQ-32B | 92.7% | 95.5% (+2.8%) | 97.3% (+4.6%) | **60.0**% |

Table 2: **Hybrid model performance on MATH500.** Results show accuracy percentages for base models, hybrid models (base + steering vectors), and thinking models. Performance improvements over base model are shown in parentheses next to hybrid and thinking model results.

| Base Model | Thinking Model | Base | Hybrid | Thinking | Gap Recovery |
|---|---|---|---|---|---|
| Llama-3.1-8B | DeepSeek-R1-Distill-Llama-8B | 30.5% | 31.9% (+1.4%) | 81.9% (+51.4%) | **2.8**% |
| Qwen2.5-14B | DeepSeek-R1-Distill-Qwen-14B | 60.2% | 72.3% (+12.1%) | 86.7% (+26.5%) | **45.7**% |
| Qwen2.5-32B | DeepSeek-R1-Distill-Qwen-32B | 61.2% | 78.1% (+16.9%) | 87.9% (+26.7%) | **63.3**% |
| Qwen2.5-32B | QwQ-32B | 63.4% | 84.4% (+21.0%) | 86.4% (+23.0%) | **91**% |

## 3.2 HYBRID MODEL IMPLEMENTATION

Our hybrid model combines the reasoning skills of the base model with the capacity to selectively apply steering vectors at appropriate points in the generation process. During generation, the model first computes SAE activations at each token position to identify the most active reasoning category. It then applies the corresponding steering vector to the base model's activations for the strongest activating category. To adjust the steering strength during generation, we apply the steering vector for a set of coefficients and steering windows (number of tokens before current token position to apply the steering vectors to) and select the steered token with the lowest perplexity according to the thinking model, ensuring that the thinking model is not thrown out of distribution.

This approach allows us to leverage the strengths of both models: the base model provides the fundamental capabilities, while the steering vectors derived from the thinking model guide when to deploy specific reasoning mechanisms. Importantly, this hybrid approach requires no parameter updates to the base model, providing strong evidence that the reasoning capabilities already exist in latent form within the base model. The effectiveness of our approach with only 15 distinct steering vectors (corresponding to our cluster size) rules out the alternative explanation that steering simply biases toward specific output tokens, as there is insufficient information to generate appropriate outputs across hundreds of diverse problems through token-level manipulation alone. Instead, our results suggest that steering activates latent reasoning modes or behaviors within the base model.

## 3.3 HYBRID MODEL RESULTS

We evaluate our hybrid model approach across multiple base model architectures and reasoning benchmarks to demonstrate the generalizability of our findings. The combinations of base and thinking models we use for our experiments are listed in Tables 1 and 2. We evaluate performance

on two mathematical reasoning benchmarks of increasing difficulty: GSM8K (Cobbe et al., 2021) for grade-school math problems and MATH500 (Hendrycks et al., 2021) for competition-level mathematics.

As shown in Tables 1 and 2, our hybrid approach demonstrates substantial performance improvements across different model architectures on GSM8K and MATH500. The hybrid model recovers 64% of the performance gap for Llama-3.1-8B on GSM8K and an impressive 91% for Qwen2.5-32B when paired with QwQ-32B on MATH500. Two interesting cases are worth noting: the 266.7% gap recovery in Table 1 for Qwen2.5-32B is surprising since it means that the hybrid model is able to perform even better than the thinking model, although the near-ceiling base accuracy makes it hard to interpret. On the other hand, the 2.8% for Llama-3.1-8B on MATH500 (Table 2) might indicate that smaller models may have less clean steering directions, yielding only marginal gains.

The results on bigger models though provide enough evidence that a significant portion of the thinking model's advantage comes from learning *when* to deploy existing reasoning mechanisms, rather than learning entirely new capabilities. See Figure 3 in Appendix C for an illustrative example of the hybrid model in action. Detailed statistics on steering vector usage patterns and the most frequently activated reasoning mechanisms for each model configuration are provided in Appendix D.

### 3.4 HYBRID MODEL ABLATION STUDIES

To assess the hybrid model's components, we ablate three factors: the specificity of the learned steering vectors, the timing of their application, and the contribution of the bias vector. We run these ablations on Qwen2.5-32B as the base with QwQ-32B (RLVR) as the thinking model on MATH500, the setting with the strongest hybrid performance (Table 2, 91% gap recovery):

- **Only-bias:** Uses only the general bias vector for steering, without any category-specific steering vectors
- **Random-firing:** Randomly selects which reasoning category to activate at each token, bypassing the SAE oracle
- **Random-vectors:** Uses random unit vectors instead of the trained steering vectors, maintaining correct dimensionality

The results are concise: *Only-bias* achieves 77.5%, indicating the bias helps but category-specific steering vectors are necessary; *Random-firing* reaches 78.3%, showing that proper timing of activation is crucial; *Random-vectors* achieves 77.8%, confirming that the learned directions are specific rather than generic. Together, these findings support that effectiveness comes from specific learned directions applied with correct, category-timed activation, consistent with our claim that thinking models primarily learn *when* to deploy reasoning mechanisms.

## 4 RELATED WORK

Some recent studies derive LLM reasoning taxonomies manually from cognitive strategies. Gandhi et al. (2025) identify four behaviors (verification, backtracking, subgoal setting, backward chaining) shared by expert humans and strong LLMs, showing these behaviors improve RL-based self-improvement and that priming with them boosts fine-tuning performance.

Other works derive taxonomies empirically. Marjanović et al. (2025) introduce a "thoughtology" of DeepSeek-R1, analyzing reasoning building blocks across chain length and cognitive style; they find an optimal chain length and that excessive rumination on initial hypotheses hinders exploration. Gema et al. (2025) corroborate that longer traces exhibit inverse-scaling performance, with similar test-time scaling observed across reasoning models (Muennighoff et al., 2025). Similarly, Sun et al. (2025) document a "ladder" of reasoning styles: moving from easy to medium tasks requires more structure, yet even extensive fine-tuning yields diminishing returns on harder problems.

A complementary line asks whether base models contain latent reasoning. Zhao et al. (2025) find RL post-training primarily amplifies pretraining patterns rather than teaching new skills. Wang et al. (2025) show that one carefully chosen example can markedly improve reasoning, suggesting minimal intervention can unlock latent reasoning "circuits". Venhoff et al. (2025) identify interpretable activation vectors corresponding to reasoning behaviors in thinking models, which can increase or decrease these mechanisms without additional fine-tuning. Ward et al. (2025) discover directions in

base models that steer reasoning behavior in thinking LLMs. To our knowledge, no prior work has systematically discovered steering vectors that induce reasoning directly in the base model itself.

On distillation, Baek & Tegmark (2025) identify feature directions in distilled models that steer different thinking modes, and Galichin et al. (2025) use Sparse Autoencoders to interpret reasoning features and enhance capabilities. Concurrently, Bogdan et al. (2025) support sentence-level decomposition of chain-of-thought, showing sentences serve distinct functions. Bridging base and reasoning-specialized models, Jia et al. (2025) integrate a learned latent action space to guide RL fine-tuning, and Zhang et al. (2025) survey reasoning-centric LLMs, emphasizing high-quality reasoning data and hybrid training.

Inference-time guidance methods aim to steer generation without fine-tuning. Li et al. (2025) propose Budget Guidance, which softly modulates token probabilities using a predictor over the remaining thinking length to meet a target budget of thinking tokens. Similarly, Fei et al. (2025) propose Nudging, which frames guided decoding as inference-time alignment by nudging next-token distributions toward a desired guidance signal.

Our work contributes an unsupervised taxonomy of reasoning mechanisms in LLMs and shows how base models can be steered along these dimensions. We unify taxonomy-driven understanding and activation-level control, supporting the view that base models possess nascent reasoning behaviors which can be selectively activated through targeted training or steering.

## 5 CONCLUSION

This work provides a novel perspective on the nature of reasoning in large language models. Through unsupervised clustering, we derived a taxonomy of reasoning steps that decompose the complex chain-of-thought processes in thinking LLMs. This interpretable taxonomy of distinct reasoning behaviors offers a framework for understanding how these models approach problem-solving tasks.

Our most significant finding is that these **reasoning behaviors are not unique to thinking models; they also exist latently within base models**. By identifying and applying the appropriate steering vectors, we demonstrated that base models can execute the same reasoning patterns when properly guided. The fact that our hybrid approach recovered up to $91\%$ of the performance gap between base and thinking models without any gradient updates provides compelling evidence for our hypothesis.

These results suggest that the reinforcement learning with verifiable rewards (RLVR) used to train thinking models primarily teaches them when to activate pre-existing capabilities rather than developing fundamentally new reasoning skills. This points to a crucial decomposition of reasoning into two components: the decision of which mechanisms to execute, and their actual execution. Thinking models excel at the former, orchestrating cognitive mechanisms already present in their base counterparts, using the additional inference-time compute to better navigate the problem space.

Our findings have important implications for future model development:

- They explain why knowledge distillation and RLVR are particularly effective for transferring reasoning capabilities to smaller models: the distillation process may primarily be teaching smaller models *when* to deploy various reasoning strategies.
- They suggest that efficient reasoning might be achievable through more targeted interventions in activation space rather than comprehensive parameter updates.
- They provide a framework for understanding and potentially addressing specific reasoning failures in LLMs by identifying and strengthening particular reasoning components.

In future work, we plan to conduct a comprehensive case study comparing the best taxonomies across different models to identify universal versus model-specific reasoning mechanisms. We also aim to develop qualitative examples and case studies demonstrating where steering significantly changes base model behavior, as well as failure cases where the hybrid model approach breaks down. We will also investigate whether this framework can be extended to induce novel reasoning capabilities in models, better understand the limitations of our current steering approach, and explore how these reasoning mechanisms develop during pre-training and fine-tuning. Ultimately, this research reframes our understanding of what makes thinking models effective and offers a path toward more efficient and targeted approaches for enhancing reasoning capabilities in language models.

## REPRODUCIBILITY STATEMENT

To ensure full reproducibility of our results, we provide comprehensive implementation details and resources. Our complete codebase is available at our GitHub repository[2], including: (1) **SAE training and evaluation code** with exact hyperparameters, layer specifications, and dictionary sizes used for all experiments; (2) **Steering vector optimization implementation** with training procedures, loss functions, and convergence criteria detailed in Appendix C; (3) **Complete prompt templates** for category generation, consistency evaluation, completeness scoring, and independence assessment used in our taxonomy evaluation pipeline; (4) **Dataset processing scripts** with exact train/validation splits, random seeds, and preprocessing steps for MMLU-Pro reasoning traces; (5) **Hybrid model evaluation framework** including the dynamic steering application logic and perplexity-based selection mechanisms; (6) **One-command reproduction scripts** that reproduce all key experimental results from SAE training through final performance evaluation. All experimental configurations, model checkpoints, and evaluation datasets are documented with version control to ensure exact reproducibility across different computational environments.

## STATEMENT ON AI-ASSISTED TOOL USAGE

This work was enhanced through the use of AI-based tools, including ChatGPT (chatgpt.com), Claude (claude.ai), and various models integrated within the Cursor IDE (cursor.com). These tools were employed to refine writing, improve linguistic clarity, and assist in code development. Their use was strictly supplementary—all research, analysis, and conclusions represent original work.

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

## Table of Contents For The Main Paper & Appendix

## A  SAE TRAINING DETAILS

Given an input vector $x \in \mathbb{R}^d$ from the residual stream and $n$ latent dimensions, a Top-K SAE learns two mappings, an encoder $f_{\text{enc}}$ and a decoder $f_{\text{dec}}$, such that:

$$z = \text{TopK}(W_{\text{enc}}(x - b_{\text{enc}})) \tag{1}$$
$$\hat{x} = W_{\text{dec}}z + b_{\text{dec}} \tag{2}$$

where $W_{\text{enc}} \in \mathbb{R}^{n \times d}$, $b_{\text{enc}} \in \mathbb{R}^n$, $W_{\text{dec}} \in \mathbb{R}^{d \times n}$, and $b_{\text{dec}} \in \mathbb{R}^d$. The training loss is then defined by the reconstruction error:

$$\mathcal{L} = \|x - \hat{x}\|_2^2 \tag{3}$$

We train our Top-K SAEs using a configuration with top-k activation sparsity where $k = 3$, meaning only the top 3 features are allowed to activate for each input. We auto-select the learning rate using the $1/\sqrt{d}$ scaling law from TinySAE (Engels, 2024): $\text{lr} = 2 \times 10^{-4}/\sqrt{n/2^{14}}$ where $n$ is the dictionary size (number of clusters), with Adam as the optimizer. Training is conducted with a batch size of 512 for a maximum of 300 epochs, implementing early stopping with a patience of 10 epochs to prevent overfitting. We apply decoder normalization after each training step, following the TinySAE implementation (Engels, 2024).

The SAEs are trained on sentence-level activations extracted from reasoning traces. We determine sentence boundaries using punctuation-based heuristics (periods, question marks, exclamation marks) and average token-level activations within each identified sentence to obtain sentence-level representations. The training data consists of 12,102 prompts from MMLU-Pro (Wang et al., 2024), which translates into 430,122 sentences of reasoning traces from the target thinking models, where we extract activations at specific layers (6 evenly distributed layers across the model depth) and use these averaged sentence activations as inputs to the SAE training process.

The specific layers used for each model are:

- **DeepSeek-R1-Distill-Llama-8B** (32 total layers): 6, 10, 14, 18, 22, 26
- **DeepSeek-R1-Distill-Qwen-1.5B** (28 total layers): 4, 8, 12, 16, 20, 24
- **DeepSeek-R1-Distill-Qwen-14B** (48 total layers): 8, 14, 20, 26, 32, 38
- **DeepSeek-R1-Distill-Qwen-32B** (64 total layers): 9, 18, 27, 36, 45, 54
- **QwQ-32B** (64 total layers): 9, 18, 27, 36, 45, 54

## B  DETAILS OF TAXONOMY EVALUATION

### B.1  CLUSTER TITLE & DESCRIPTION GENERATION

We use OpenAI's o4-mini model to generate the cluster title and description.

Concretely, we prompt this model to carefully look at the examples and identify the shared reasoning strategy or cognitive mechanism, common linguistic patterns or structures, specific phrases or words common to the category, and the functional role within the overall reasoning process. The model

then produces a concise title naming the specific reasoning function and a detailed description that explains what the function does, what is included, and what is excluded from this category. This prompt is shown below:

```
1  Analyze the following [N] sentences from an LLM reasoning trace. These sentences are
       grouped into a cluster based on their similar role or function in the reasoning
       process.
2
3  Your task is to identify the precise cognitive function these sentences serve in the
       reasoning process. Consider the reasoning strategy or cognitive operation being
       performed.
4
5  Sentences:
6  '''
7  [LIST OF EXAMPLE SENTENCES]
8  '''
9
10 Look for:
11 - Shared reasoning strategies or cognitive mechanisms
12 - Common linguistic patterns or structures
13 - Functional role within the overall reasoning process
14
15 [OPTIONAL: CATEGORY EXAMPLES SECTION WITH 5 EXAMPLE CATEGORIES]
16
17 Your response should be in this exact format:
18 Title: [crisp, single-concept title without slashes, parentheses, or compound phrases]
19 Description: [3-4 sentences explaining (1) the specific reasoning process this cluster
       represents, (2) what is INCLUDED in this category, (3) what is NOT INCLUDED in this
       category]
20
21 Guidelines for titles:
22 - Use simple, clear nouns or verb phrases
23 - Avoid slashes (/) and parentheses ()
24 - Capture one core reasoning concept
25
26 Guidelines for descriptions:
27 - Focus on the specific cognitive or reasoning function
28 - Avoid abstracting too much from the specific examples
29 - Mention specific phrases or words that are common in the examples
30 - Be precise enough that someone could reliably identify new examples of this reasoning
       function.
31
32 In summary, the description should be as sharp and specific as possible, and the title
       should be as simple and abstract as possible.
```

Prompt 1: Prompt used for generating cluster descriptions and titles

## B.2 CONSISTENCY (F1 SCORE)

To evaluate how well our categories can reliably classify individual sentences, we implement a binary classification task. For each category, we sample example sentences from within the category (positive examples) and from outside the category (negative examples). An LLM-based autograder (OpenAI's GPT-4.1-mini) receives the category title and description along with these examples and must classify each as either belonging to the category or not. We calculate precision, recall, and F1 scores for each category, then take the average F1 score across all categories as our overall consistency score. The complete prompt is shown below:

```
1  # Task: Binary Classification of Reasoning Sentences by Function
2
3  You are an expert at analyzing the *function* of sentences within a longer chain of
       reasoning. Your task is to determine if each sentence below performs the specific
       cognitive or procedural role described.
4
5  **Core Principle:** Do not focus on the surface-level topic of the sentence. Instead,
       abstract away from the specific content and ask: "What *job* is this sentence doing in
        the reasoning trace?"
6
7  ## Category Description:
8  Title: [TITLE]
9  Description: [DESCRIPTION]
10
11 ## Sentences to Classify:
12 [FORMATTED SENTENCES]
13
14 ## Instructions:
15 1. For each sentence, identify its functional role in a potential reasoning process.
```

```
16 2. Compare this role to the category description provided.
17 3. If the sentence's function matches the description, assign "Yes". Importantly, a
        sentence might not match a description word-for-word, but it might serve the same
        underlying purpose.
18 4. If the sentence's function does not align with the category, assign it "No".
19 5. Respond with "Yes" or "No" for each sentence.
20
21 ## Response Format:
22 Your response must follow this exact JSON format:
23 ```json
24 {
25   "classifications": [
26     {
27       "sentence_id": <sentence idx>,
28       "belongs_to_category": "Yes" or "No",
29       "explanation": "Brief explanation of your reasoning"
30     }
31   ]
32 }
33 ```
34
35 Only include the JSON object in your response, with no additional text before or after.
```

Prompt 2: Prompt used for the F1 score (accuracy) autograder

## B.3 COMPLETENESS (CONFIDENCE SCORE)

We evaluate how well individual sentences fit their assigned categories by having an LLM (GPT-4.1-mini) rate the quality of each assignment on a scale from 0-10. This measures the confidence in our category assignments and serves as our completeness metric. The scores are afterwards normalized to a 0-1 scale for compatibility with the final score calculation. See Appendix B.3 for the complete prompt.

```
1 You are an expert at analyzing how well individual sentences match their assigned reasoning
        function categories. Your task is to evaluate how well a given sentence exemplifies
        the specific cognitive or procedural role described in its assigned category.
2
3 # Sentence to Evaluate:
4 [SENTENCE]
5
6 # Assigned Category:
7 Title: [TITLE]
8 Description: [DESCRIPTION]
9
10 # Instructions:
11 1. Carefully analyze the sentence's content and the functional role it might display in a
        reasoning process.
12 2. Compare this content and role to the category description provided.
13 3. Consider how well the sentence matches the category description.
14 4. Provide a brief explanation of your reasoning.
15 5. Rate the fit on a scale from 0-10, where:
16   - 0 = Very poor fit, sentence does not match the category at all
17   - 10 = Perfect fit, sentence matches exactly the category description
18
19 # Response Format:
20 Your response must follow this exact JSON format. The explanation must be a single-line
        string with no newlines:
21 ```json
22 {
23   "explanation": "Brief explanation of how well the sentence matches the category and your
        reasoning for the score",
24   "completeness_score": <integer from 0-10>
25 }
26 ```
27
28 Only include the JSON object in your response, with no additional text before or after.
```

Prompt 3: Prompt used for the completeness autograder

## B.4 INDEPENDENCE (SEMANTIC ORTHOGONALITY)

To ensure that our taxonomy categories represent functionally distinct reasoning mechanisms, we evaluate the semantic similarity between all pairs of categories using an LLM-based approach. For each pair of categories in a cluster, an LLM (GPT-4.1-mini) evaluates how similar they are in terms

of their underlying cognitive or functional purpose on a scale from 0-10, where 0 means completely different reasoning functions and 10 means essentially the same function. We then calculate the semantic orthogonality score as the fraction of category pairs that have an orthogonality score above a threshold (0.5), where orthogonality is defined as $1 - \text{similarity}$, indicating functional independence between categories. The complete prompt is shown below:

```
1  # Task: Semantic Similarity Evaluation
2
3  You are an expert at analyzing the semantic similarity between different reasoning
        functions. Your task is to evaluate how similar two categories of reasoning sentences
        are in terms of their underlying cognitive or functional purpose.
4
5  ## Category 1:
6  Title: [TITLE1]
7  Description: [DESCRIPTION1]
8
9  ## Category 2:
10 Title: [TITLE2]
11 Description: [DESCRIPTION2]
12
13 ## Instructions:
14 Rate the semantic similarity between these two categories on a scale from 0 to 10, where:
15 - 0 = Completely different reasoning functions
16 - 5 = Somewhat related but distinct functions
17 - 10 = Essentially the same reasoning function, just described differently
18
19 Consider:
20 1. The underlying cognitive process or reasoning operation
21 2. The functional role within a reasoning trace
22 3. Whether sentences from one category could reasonably belong to the other
23
24 Focus on functional similarity rather than surface-level word overlap.
25
26 ## Response Format:
27 Your response must follow this exact JSON format:
28 ```json
29 {
30   "explanation": "Brief explanation of your reasoning for this score",
31   "similarity_score": <integer from 0-10>
32 }
33 ```
34
35 Only include the JSON object in your response, with no additional text before or after.
```

Prompt 4: Prompt used for the semantic orthogonality evaluation

### B.5 DECODER WEIGHT VECTOR ORTHOGONALITY

The independence of our taxonomy can also be measured by the orthogonality between the decoder latents (centroids) in our Sparse Autoencoder (SAE). For a set of decoder weight vectors $\{w_1, w_2, ..., w_n\}$ where $n$ is the number of categories, we calculate:

$$\text{Orthogonality}_{i,j} = \frac{w_i \cdot w_j}{||w_i|| \cdot ||w_j||} \tag{4}$$

This produces a cosine similarity matrix where values close to 0 indicate nearly orthogonal (independent) features. We then compute:

- The average absolute cosine similarity between all pairs of latents
- The maximum absolute cosine similarity between any pair of latents

Lower values for both metrics indicate better independence between our taxonomy categories. This orthogonality analysis ensures that our categories represent distinct reasoning mechanisms rather than variations of the same underlying process.

However, in practice, we found that these cosine similarity values were consistently very high (near 1.0) across different SAE configurations, providing limited discriminative power for comparing different taxonomies. This led us to adopt the semantic orthogonality metric instead, which better captures functional distinctness between reasoning categories.

### B.6 Choice of LLM Models for Taxonomy Evaluation

We employ different LLM models for different evaluation tasks based on their computational re-
quirements and criticality to downstream performance. Category title and description generation
uses OpenAI's o4-mini, a more sophisticated reasoning model, because these titles fundamentally
determine the semantic boundaries of each category and directly impact all subsequent evaluation
metrics. In contrast, consistency, completeness, and independence evaluations use GPT-4.1-mini, a
capable but more cost-effective model, as these tasks involve more straightforward classification and
rating given well-defined categories. This design choice is further motivated by our evaluation scale:
we generate 5 repetitions of category titles for each configuration across our extensive grid search,
making the computational cost of using premium models for all evaluation steps prohibitive while
maintaining evaluation quality where it matters most.

### B.7 Scoring Normalization

For our grid search visualization and comparison across different configurations, we normalize
each metric to a 0-1 scale using min-max normalization within each model. This normalization
is performed across all layer and cluster size combinations for a given model, ensuring that the
final normalized score reflects relative performance within each model's configuration space. The
normalization formula is:

$$\text{Normalized Score} = \frac{\text{Raw Score} - \text{Min Score}}{\text{Max Score} - \text{Min Score}} \tag{5}$$

where Min Score and Max Score are computed across all layer/cluster combinations for a single
model.

## C Details of Hybrid Model Evaluation

### C.1 Example Selection and Training Procedure

A steering vector $v \in \mathbb{R}^d$ can be identified by comparing the activations of a model in two different
states, for example, when the model is generating factual versus misleading information. The
difference between these activation states forms a vector that can then be used to steer the model's
behavior in the desired direction:

$$v = \mathbf{E}[h(x_{\text{target}}) - h(x_{\text{base}})] \tag{6}$$

where $h(x)$ represents the hidden activations at a particular layer when processing input $x$.

For each reasoning category, we select the $8192$ sentences with the highest SAE activation and
compute the base model's perplexity on those sentences, given the previous rollout. Then we select
the top $2048$ highest-perplexity sentences as training examples. This balances selecting examples that
strongly represent the category while encouraging examples that the base model would find highly
improbable, which are exactly the examples that are most useful for steering vector optimization.

We train the steering vectors for up to $50$ iterations with a learning rate of $1\mathrm{e}{-2}$ (with cosine
scheduler) and minibatch size of 6, using activation perplexity selection to choose optimal examples
during training. To prevent overfitting, we implement early stopping with a minimum delta of
$0.01$ and patience of 5 steps. The optimization objective uses standard next token prediction loss
(cross-entropy) computed only on the target completion tokens, excluding the prompt tokens from
the loss calculation.

### C.2 Prompt Template

All training examples for steering vector optimization share a common prefix structure to induce
step-by-step reasoning behavior in the base model, which has not been fine-tuned for instruction
following:

Figure 3: **Hybrid model in action.** Example of a hybrid model (Qwen2.5-32B as the base model with steering vectors trained on QwQ-32B thinking model) solving a MATH500 problem, showing how steering vectors are dynamically applied based on SAE activations to guide the base model's reasoning process. The model successfully identifies and applies appropriate reasoning mechanisms at each step, demonstrating the effectiveness of our approach in practice.

```
Task: Answer the question below. Explain your reasoning step by step.

Question: [original question]

Step by step answer: [thinking process]
```

## D    STEERING VECTOR USAGE STATISTICS

This section provides detailed statistics on the application of steering vectors across different model configurations and reasoning benchmarks. These statistics reveal which reasoning mechanisms are most frequently deployed and their contribution to overall performance improvements.

### D.1    LLAMA-3.1-8B WITH DEEPSEEK-R1-DISTILL ON GSM8K

For the Llama-3.1-8B model with DeepSeek-R1-Distill steering on GSM8K, steering vectors were applied to an average of 125.96 tokens per problem out of 1641.56 total tokens, representing 7.7%

of the generation process. The most frequently activated reasoning mechanism was "Planning Next Steps" (18.7% of steered tokens), followed by "Confirming Reasoning Steps" (16.2%) and "Numeric Calculation Steps" (13.8%). The majority of steering applications used a coefficient of 0.3 (68.6%), indicating that relatively gentle nudges were most effective for this configuration.

### D.2    LLAMA-3.1-8B WITH DEEPSEEK-R1-DISTILL ON MATH500

On the more challenging MATH500 benchmark, the same model configuration showed different steering patterns. Steering was applied to 91.55 tokens per problem on average out of 1114.06 total tokens (8.2% steered fraction). "Planning Next Steps" remained the most used mechanism (21.0%), and "Mathematical Computation Steps" was equally prominent (21.0%), reflecting the increased mathematical complexity. The steering coefficient distribution shifted toward 0.5 (85.2%), suggesting that stronger interventions were needed for these harder problems.

### D.3    QWEN2.5-14B WITH DEEPSEEK-R1-DISTILL ON GSM8K

The Qwen2.5-14B configuration showed a markedly different pattern, with steering applied to 45.49 tokens per problem out of 738.91 total tokens (6.2% steered fraction). The steering was dominated by "Problem Restatement" (55.7%) and "Metacognitive Markers" (41.7%), with minimal use of computational reasoning mechanisms.

### D.4    QWEN2.5-14B WITH DEEPSEEK-R1-DISTILL ON MATH500

On MATH500, the same model maintained a similar pattern with 62.33 steered tokens per problem out of 823.96 total tokens (7.6% steered fraction). "Problem Restatement" remained dominant (67.0%), followed by "Metacognitive Markers" (29.8%). The steering coefficient remained primarily at 0.5 (77.3%).

### D.5    QWEN2.5-32B WITH DEEPSEEK-R1-DISTILL ON GSM8K

The largest model configuration with DeepSeek-R1-Distill showed relatively low steering intensity, applying vectors to 32.02 tokens per problem out of 622.16 total tokens (5.1% steered fraction). The most frequent mechanism was "Evaluating Expressions" (29.8%), followed by "Drawing Conclusions" (23.6%). The coefficient distribution favored 0.5 (67.6%).

### D.6    QWEN2.5-32B WITH DEEPSEEK-R1-DISTILL ON MATH500

On MATH500, this configuration steered 54.18 tokens per problem out of 800.09 total tokens (6.8% steered fraction). "Evaluating Expressions" became even more dominant (43.3%), with "Computational step initiation" (17.5%) and "Drawing Conclusions" (17.2%) also playing significant roles. The steering coefficient remained primarily at 0.5 (74.4%).

### D.7    QWEN2.5-32B WITH QWQ-32B ON GSM8K

The QwQ-32B thinking model produced a different steering pattern, with 30.82 steered tokens per problem out of 300.96 total tokens (10.2% steered fraction). "Numeric computations" was the most frequent mechanism (28.9%), followed by "Stating Known Equations" (28.2%). The coefficient distribution was heavily concentrated at 0.5 (88.3%).

### D.8    QWEN2.5-32B WITH QWQ-32B ON MATH500

On the challenging MATH500 benchmark, this configuration showed higher steering intensity with 98.08 steered tokens per problem out of 920.49 total tokens (10.7% steered fraction). The mechanism distribution was balanced, with "Stating Known Equations" (24.3%), "Presenting Conclusions" (21.3%), "Numeric computations" (20.7%), and "Planning Next Steps" (20.5%) all playing substantial roles. The steering coefficient remained primarily at 0.5 (69.7%), with increased use of higher coefficients (0.9: 8.4%, 0.8: 6.8%).

These statistics reveal that steering vector usage patterns are dependent on both base model capabilities and task complexity. Smaller models require more frequent and diverse steering interventions, while larger models benefit from targeted steering of specific reasoning mechanisms. The coefficient distributions suggest that optimal steering strength varies by model size and difficulty, with gentle interventions often being most effective.

# E SPARSE AUTOENCODER FEATURES

In this section, we provide detailed tables showing the complete reasoning taxonomies for our best-performing SAE configurations. For transparency and to demonstrate the full scope of our approach, we present all discovered categories rather than a curated subset. For each model, we list all category titles and representative examples for the sparse autoencoder features identified during our analysis.

## E.1 DEEPSEEK-R1-DISTILL-LLAMA-8B (LAYER 6, DICT SIZE 15)

Table 3: Categories and representative examples for DeepSeek-R1-Distill-Llama-8B (Layer 6, Dict Size 15)

| Category | Representative Example |
|---|---|
| Recalling Mathematical Formulas | "The formula for heat transfer through conduction in a cylindrical or spherical object is given by Q = (k * A * $\Delta$T * t) / d, where Q is the heat transferred, k is the thermal conductivity, A is the surface area, $\Delta$T is the temperature difference, t is time, and d is the thickness." |
| Retrieving Factual Knowledge | "I think C3 refers to a type of photosynthesis where carbon fixation happens in the stroma of the chloroplast, and C4 is another type where it happens in a specialized cell structure called the bundle sheath." |
| Listing considerations | "I also think about the concept of market saturation." |
| Conditional Outcome Projection | "So, the son's argument would be that the covenant is enforceable against him because it's a real covenant that runs with the land, and the neighbor can enforce it. The son didn't have a valid defense because he didn't record and didn't know, but knowledge isn't necessary for real covenants." |
| Conditional Causal Reasoning | "I think the key point is that when both aggregate supply and aggregate demand increase, the price level might decrease because the demand is pulling it down, but the supply is also increasing, which might make the price level not decrease as much as it would if only demand was increasing." |
| Drawing Conclusions | "So, putting it all together, I think Aristotle's philosophy is the most consistent with the idea of three major life tasks because his teachings on virtue and the structure of a good life include these areas as important components." |
| Restating Given Data | "A 0.1 m diameter, 0.1 m high solid copper cylinder is initially at 180°C. It is then placed in a room and is allowed to cool to a final temperature of 30°C. Assuming copper to have a density of 8954 kJ/kg·°K, calculate the heat transfer and the irreversibility of the process if the temperature of the surroundings (T_0) is 25°C." |
| Verifying Intermediate Steps | "Wait, but hold on, is that correct?" |
| Confirming Reasoning Steps | "So that seems correct." |
| Mathematical Computation Steps | "u'(t) = (1/6) e^{}(t/6) (C1 cos($\beta$ t) + C2 sin($\beta$ t)) + e^{}(t/6) [ -C1 $\beta$ sin($\beta$ t) + C2 $\beta$ cos($\beta$ t) ]." |
| Numeric Calculation Steps | "First, 4 * 1.60218 = 6.40872, and 0.4781 * 1.60218 $\approx$ 0.4781 * 1.6 $\approx$ 0.76336, so total $\approx$ 6.40872 + 0.76336 $\approx$ 7.17208 x 10^{}-16 kJ." |
| Task Formulation | "Okay, so I'm trying to figure out what the Supreme Court ruled regarding older workers and job discrimination suits." |

Table 3: Categories and representative examples for DeepSeek-R1-Distill-Llama-8B (Layer 6, Dict Size 15)

| Category | Representative Example |
| --- | --- |
| Proposing Possibilities | "Realists might have been more pragmatic, advising governments on how to navigate a dangerous world through strength and alliances, while peace researchers might have pushed for more proactive measures to address the root causes of conflict, like economic policies or social justice issues that could lead to tensions." |
| Planning Next Steps | "Let me write down the equations step by step." |
| Expressing Uncertainty | "But I'm not sure about the specifics, so I'll have to stick to general principles." |

## E.2 DEEPSEEK-R1-DISTILL-QWEN-1.5B (LAYER 4, DICT SIZE 25)

Table 4: Categories and representative examples for DeepSeek-R1-Distill-Qwen-1.5B (Layer 4, Dict Size 25)

| Category | Representative Example |
| --- | --- |
| Recalling Constants and Parameters | "The gas is N2O, so I need to find the moles of N2O produced and then convert that to volume at 1092°C. Wait, but the temperature is given as 1092°C. I remember that gas volumes are usually calculated at standard temperature and pressure (STP), which is 0°C (273 K) and 1 atm." |
| Listing Problem Facts | "Ash wants to set up a savings account for her daughter's education, and she needs to figure out how much she has to deposit annually for 17 years at a 5% interest rate to reach a total of $20,000." |
| Stating Conclusions | "So, the ratio is zero." |
| Drawing Conclusions | "So, if the teres minor is the most common, then the answer is that the teres minor is the most likely, but the question is phrased as "which of the following tendons," so maybe the answer is that the teres minor is the most likely, but the question is testing whether the person knows that the teres minor is the most common, so the answer is the teres minor, but the question is phrased as "which of the following tendons," so perhaps the answer is that the teres minor is the most likely, but the question is a bit confusing." |
| Intermediate Arithmetic Calculations | "$233.40 * 0.0086 = 233.40 * 0.008 + 233.40 * 0.0006 = 1.8672 + 0.14004 = 1.8672 + 0.14004 = 1.8672 + 0.14004 = 2.00724$." |
| Recalling Domain Knowledge | "So, for an isothermal process, the work done on the gas is equal to the heat exchanged, but since it's an ideal gas, the internal energy depends only on temperature, which is constant, so the work done on the gas is equal to the change in internal energy, which is zero." |
| Hypothetical Elaboration | "He might have argued that restricting the use of certain languages could limit students' ability to participate in the broader social and cultural life, which is important for their personal and societal development." |
| Self-correction cues | "Wait, but wait a second." |
| Speculating Alternatives | "Maybe I need to assume that the heater is at a certain temperature, or perhaps it's a blackbody?" |
| Stating Given Information | "- **$F_2$ ($F_1 \times F_1$)**: Mean = 30 mm, Variance = 5.10 mm$^2$, Environment = Strain A" |

Table 4: Categories and representative examples for DeepSeek-R1-Distill-Qwen-1.5B (Layer 4, Dict Size 25)

| Category | Representative Example |
|---|---|
| Elementary Deduction | "So, the area where x < y is the area above the line y=x from (0,0) to (2,2), and then from (2,2) to (3,2), it's a rectangle where y can be anything from 2 down to 0, but since y is already 2, x can be from 0 to 3, but wait, no, because y is fixed at 2, so x can be from 0 to 3, but in this region, y is fixed at 2, so x < y would mean x < 2." |
| Metacognitive Prompts | "Let me think." |
| Expressing mathematical relationships | "Q = (2 * π * L) * (T(r1) - T(r2)) / (k_avg * ln(r2 / r1))" |
| Planning calculation steps | "I need to solve this system of equations to find the values of a, b, and c. Once I have those, I can plug in t = -1 into the polynomial to find f(-1)." |
| Hypothesis Generation | "Myo-invasive heart disease is a type of heart disease where the heart muscle is damaged and can't function properly, leading to symptoms like ptosis, weakness, and difficulty rising from a chair." |
| Rule Application | "Wait, but the statute of limitations was set so that the claim against Carla expired the day before Ann's lawsuit. So, if Bea's agreement was unenforceable, then she can't sue Carla for the debt, but she can still sue Ann for the clubs." |
| Background Knowledge Retrieval | "I remember that in electrical circuits, there's something called the force-voltage analogy, which is used to model mechanical systems using electrical components." |
| Question Focus Clarification | "But the question is asking for the most reasonable conclusion, not necessarily about the student's performance." |
| Planning Next Steps | "First, let's understand the current situation." |
| Stepwise Decomposition | "Hmm, let me try to break this down step by step." |
| Problem Restatement and Next-Step Planning | "Okay, so I'm trying to understand this statement: "______ is an employee's preferred ratio between work-related and non-work-related activities which, due to intensification of work and technological shifts, has become a hotly contested issue in recent years." |
| Acknowledging Uncertainty | "But I'm not entirely sure about this." |
| Considering Additional Factors | "I should also think about the possible causes again." |
| Validating Intermediate Conclusions | "That seems correct." |
| Causal Conditionals | "Total revenue is price times quantity, so if the price of the product is fixed, then total revenue would be P*Q. But if the price is fixed, then increasing the wage rate would increase the cost, but the revenue might not change because the price is fixed." |

### E.3 DEEPSEEK-R1-DISTILL-QWEN-14B (LAYER 38, DICT SIZE 5)

Table 5: Categories and representative examples for DeepSeek-R1-Distill-Qwen-14B (Layer 38, Dict Size 5)

| Category | Representative Example |
|---|---|
| Numeric Computation | "51 * 4 = 204, 51 * 0.184 = approximately 9.4, so total is 204 + 9.4 = 213.4 kJ/K. Then, 213.4 * 2.5 = 533.5 kJ." |

Table 5: Categories and representative examples for DeepSeek-R1-Distill-Qwen-14B (Layer 38, Dict Size 5)

| Category | Representative Example |
|---|---|
| Conditional Reasoning | "The Due Process and Equal Protection arguments are weaker because the emergency nature of the bill might justify some procedural issues, and there's no clear indication of unequal treatment unless the tax is applied differently to in-state and out-of-state entities, which isn't explicitly stated." |
| Problem Restatement | "Okay, so I have this problem where I need to determine the absolute zero temperature in degrees Celsius using the densities of air at three different temperatures: -85°C, 0°C, and 100°C. The densities given are 1.877 g/dm$^3$, 1.294 g/dm$^3$, and 0.946 g/dm$^3$ respectively." |
| Metacognitive Markers | "Hmm, I'm a bit rusty on this, but let me try to think it through." |
| Recall of Domain Knowledge | "From what I remember, constructivism is a learning theory that suggests people construct their own understanding and knowledge through their experiences and by reflecting on those experiences." |

## E.4 QwQ-32B (Layer 27, Dict Size 10)

Table 6: Categories and representative examples for QwQ-32B (Layer 27, Dict Size 10)

| Category | Representative Example |
|---|---|
| Adjusting problem framing | "Alternatively, maybe the question is structured so that the options include both, but since the user hasn't given options, I have to go with the standard answer." |
| Brainstorming Additional Aspects | "Also, maybe mention psychological theories explicitly, like Skinner's operant conditioning, Pavlov's classical conditioning, and maybe Maslow's hierarchy where esteem and social belonging are needs that verbal stimuli can address." |
| Numeric computations | "Let me do 60 * 0.03 = 1.8, and 60 * 0.0006 = 0.036, so total is 1.8 + 0.036 = 1.836 cal/°C. Then multiply by 307°C. Hmm, 1.836 * 300 = 550.8, and 1.836 *7 = 12.852, so total is 550.8 +12.852 = 563.652 cal." |
| Proposing Causal Hypotheses | "Also, maybe the government wants to ensure that the market remains dynamic and innovative, which collusion might hinder because if companies aren't competing, they might not feel the need to improve their products or services." |
| Recalling Domain Knowledge | "Since they are parallel, maybe I can use some proportionality theorem, like Thales' theorem or the basic proportionality theorem (Thales' theorem), which states that if a line is drawn parallel to one side of a triangle intersecting the other two sides, then it divides them proportionally." |
| Drawing Deductive Inferences | "Also, the confrontation clause: since the victim is deceased, the defendant can't cross-examine them, but the dying declaration exception is an established exception to the confrontation clause as well, based on precedent like Tennessee v. Street.
So, if the court determines that the victim's statement was made in reasonable belief of impending death, then it's admissible as a dying declaration." |
| Stating Known Equations | "So, the Clausius-Clapeyron equation in the form ln(P) = -$\Delta$H_vap/(R) * (1/T) + C. The slope is -$\Delta$H_vap/R, so if I can find the slope, I can solve for $\Delta$H_vap." |
| Articulating the Next Subgoal | "Okay, so I need to figure out which of the given options definitely increases the equilibrium price of corn in a competitive market." |

Table 6: Categories and representative examples for QwQ-32B (Layer 27, Dict Size 10)

| Category | Representative Example |
|---|---|
| Planning Next Steps | "Let me think again." |
| Presenting Conclusions | "Therefore, both methods give the same answer, so I think that's correct." |

## E.5    DEEPSEEK-R1-DISTILL-QWEN-32B (LAYER 27, DICT SIZE 15)

Table 7: Categories and representative examples for DeepSeek-R1-Distill-Qwen-32B (Layer 27, Dict Size 15)

| Category | Representative Example |
|---|---|
| Proposing Explanations | "In summary, Sandel's case against moral engineering likely revolves around several key points: undermining personal autonomy and responsibility, leading to inauthentic actions, being paternalistic, causing unintended consequences, ignoring the social and cultural context, potentially failing to achieve lasting moral change, raising ethical concerns about technology use, risking overreach, and being less effective than other methods that respect individual agency." |
| Enumerating additional factors | "I should also think about the biological perspective." |
| Problem Framing | "Okay, so I need to figure out which statement correctly expresses the relationship between aging and sexual functioning." |
| Inferring Causal Effects | "This would cause the supply curve to shift to the left because they need a higher price to cover their increased costs, or they might reduce the quantity supplied at each price level." |
| Computational step initiation | "Let me plug in the numbers step by step." |
| Memory Retrieval | "HDL stands for High-Density Lipoprotein, and I think its main apoprotein is apoA-I. HDL is often called the "good" cholesterol because it helps remove cholesterol from the arteries and transport it back to the liver for excretion." |
| Verifying Intermediate Results | "Yeah, that seems correct." |
| Drawing Conclusions | "So, putting it all together, positive reinforcement is the strategy that will most effectively increase Mary's likelihood of completing her seat work in the long term." |
| Stating Given Problem Data | "The reactor has a maximum power rating of 150 W per meter of pipe, and it operates at 350 K. The flow rate is 5 kg/hr, and the water enters at 290 K. The pipe has an internal diameter of 0.005 m. I need to find the length of the pipe required for the heat transfer and the maximum exit temperature of the water." |
| Evaluating Expressions | "- The entry in the second row, first column (2,1) is 2 + 1 = 3." |
| Intermediate Numerical Computations | "Calculating that, \{}( \{}pi \{}) is approximately 3.1416, so 3.1416 * 0.000225 is about 0.000706858 square meters." |
| Recalling Scientific Laws and Formulas | "The Nusselt number (Nu) is a dimensionless number that relates the convective heat transfer coefficient to the thermal properties of the fluid and the geometry of the system." |
| Recalling Formulas | "$\lambda$ = h / (m * v_n) = h / (m * sqrt(3*k_B*T / m))" |
| Expressing Uncertainty | "I think I need to confirm this, but since I can't look it up right now, I'll go with "moral particularism" as the answer, but I'm not 100% sure." |
| Stating Legal Rules | "In this case, the shopper is suing under strict liability, so the key elements she needs to prove are: (1) the product was defective, (2) the defect caused her injury, and (3) she was using the product in a way that was foreseeable." |

