# OpenReview forum: "Base Models Know How to Reason, Thinking Models Learn When"
_ICLR.cc/2026/Conference — ICLR 2026 Conference Withdrawn Submission_

### Official Review · Reviewer_NFBd · 2025-10-23

**Soundness:** 2
**Presentation:** 2
**Contribution:** 3
**Rating:** 6
**Confidence:** 2

**Summary:**

The paper investigates what differentiates "thinking" models (like DeepSeek) from their base counterparts, claiming that the base models already contain latent reasoning mechanisms, and that e.g. Reinforcement Learning with Verifiable Rewards mostly teaches *when* to activate these mechanisms. To this end, the paper first designs an unsupervised taxonomy of reasoning mechanisms. Then, it derives steering vectors in activation space that correspond to each reasoning category, and constructs hybrid models that apply the right steering vectors to base models at the right time. Empirically, the paper shows that this steering recovers a large part of the performance gap between "thinking" models and base models on two datasets. This shows that, when properly guided, base models can execute similar reasoning patterns as "thinking" models.

**Note:** *I am not very familiar with this field, so I do not know the existing state-of-the-art. This also means I cannot evaluate well how novel the approach is compared to existing methods, nor any claims about this.*

**Strengths:**

- The hypothesis (reasoning mechanisms exist latently in base models) is clear and interesting.
- Experiments investigate the approach on 4 LLMs (Llama and 3 Qwen variants).
- The experimental results are strong, showing a large improvement in performance due to the approach.

**Weaknesses:**

- No standard-deviation or confidence intervals are reported for the results.
- The main text is quite informal. The method is explained in pure text, with the absence of even a single formal equation.
- Some details that seem quite important to understand the approach are delegated to the appendix (e.g. details of how you find and compute the steering vectors, line 280). However, in Section 3.1, you *do* seems to explain how you find steering vectors in base models. This is confusing, as in line 280 you state that you explain it in the appendix instead.

**Questions:**

1. "We measure the completeness of our categories by evaluating the confidence that an LLM has in classifying sentences in their categories" Why are these two things related (line 216)?
2. "maximizes the next token prediction loss" shouldn't this be "minimizes [...]" (line 307)? If not, why would you want to maximize a loss?
3. How many seeds (runs) did you use in your experiments?

---

### Official Review · Reviewer_HA8Q · 2025-10-24

**Soundness:** 2
**Presentation:** 3
**Contribution:** 2
**Rating:** 4
**Confidence:** 4

**Summary:**

The paper investigates the performance gap between "base" and "thinking" language models, arguing that the latter's superiority in reasoning tasks stems not from acquiring new skills, but from learning when to apply capabilities already latent in the base model. The authors introduce a novel unsupervised method using Sparse Autoencoders (SAEs) to discover an interpretable taxonomy of reasoning mechanisms. They then demonstrate that by applying mechanism-specific "steering vectors" to a base model's activations at oracle-determined moments, they can recover up to 91% of the performance gap on math benchmarks without any weight updates. The work concludes that post-training methods like RLVR primarily teach models to orchestrate, rather than execute, reasoning.

**Strengths:**

1. **Novel and Insightful Central Hypothesis:** The "learning when, not how" framework provides an elegant and powerful conceptualization of the role of reasoning-focused post-training, shifting the focus from skill acquisition to skill orchestration.
2. **Principled Unsupervised Taxonomy Discovery:** The use of SAEs to create a bottom-up, data-driven taxonomy of reasoning mechanisms is a strong methodological contribution. It offers a more objective way to decompose complex reasoning chains compared to manual annotation.
3. **Compelling Empirical Validation:** The hybrid steering model serves as a clever "proof of concept." Recovering up to 91% of the performance gap without any fine-tuning provides strong causal evidence for the existence of latent reasoning primitives within base models.

**Weaknesses:**

1. **Lack of Crucial Baseline Comparisons:** The paper evaluates its hybrid model against the base and thinking models but misses a critical, simpler baseline: standard Chain-of-Thought prompting [1] or Zero-Shot-CoT [2] applied directly to the base model. These methods also elicit reasoning without weight updates. Without this comparison, it's impossible to know how much of the performance gain comes from the sophisticated steering vector mechanism versus the simple act of prompting the model to "think step-by-step". This omission makes it difficult to assess the true added value of the proposed hybrid approach.

2. **Substantial Overlap with Recent Work:** The paper's primary empirical discover, that reasoning skills can be elicited in base models by training/applying simple steering vectors, is not entirely novel. The concurrent work [3] demonstrates a similar, and arguably more practical, result: training a steering vector via RL is sufficient to match the performance of fully fine-tuned models. This end-to-end trainable approach weakens the novelty of this paper's central finding about latent skills. Consequently, this paper's main unique contribution is narrowed down to the SAE-based interpretability framework, rather than the discovery of activatable skills itself.

3. **Reliance on an Impractical Oracle:** The hybrid model's success hinges on an external oracle (a classifier trained on the thinking model's behavior) to decide which steering vector to apply at each step. While this is an effective proof-of-concept for demonstrating latent capabilities, it is not a practical, standalone system. The paper does not address how a model might learn this orchestration policy autonomously, which is the core challenge that "thinking models" are trained to solve.

4. **Limited Empirical Scope:** The experiments are exclusively focused on mathematical reasoning, which narrows the scope of the claims about general reasoning mechanisms. It is unclear if the discovered taxonomy and the effectiveness of steering would transfer to other domains like commonsense or logical deduction. Furthermore, the significantly weaker results on the Llama-3.1-8B model for MATH500 (2.8% gap recovery) are underexplored. This discrepancy might suggest that the "knows how" hypothesis is model-dependent. As shown by Shao et al. [4], different model families (like Qwen and Llama) have strong, distinct pre-existing biases (e.g., a tendency for "code reasoning" in Qwen). The high performance recovery in Qwen models might be due to steering activating these specific, powerful priors related to math, which may be less developed in Llama models. This suggests the results could be confounded by the choice of a specific model family and a specific domain (math), rather than representing a universal principle.


[1] Wei, J., Wang, X., Schuurmans, D., Bosma, M., Xia, F., Chi, E., ... & Zhou, D. (2022). Chain-of-thought prompting elicits reasoning in large language models. Advances in neural information processing systems, 35, 24824-24837.

[2] Kojima, T., Gu, S. S., Reid, M., Matsuo, Y., & Iwasawa, Y. (2022). Large language models are zero-shot reasoners. Advances in neural information processing systems, 35, 22199-22213.

[3] Sinii, V., Gorbatovski, A., Cherepanov, A., Shaposhnikov, B., Balagansky, N., & Gavrilov, D. (2025). Steering LLM Reasoning Through Bias-Only Adaptation. arXiv preprint arXiv:2505.18706.

[4] Shao, R., Li, S. S., Xin, R., Geng, S., Wang, Y., Oh, S., ... & Zettlemoyer, L. (2025). Spurious rewards: Rethinking training signals in rlvr. arXiv preprint arXiv:2506.10947.

**Questions:**

1. While you performed a grid search for the SAE taxonomy, it's unclear how sensitive the final hybrid model's performance is to these choices. How would the "gap recovery" percentage change if you used a different number of clusters (e.g., 30 instead of 15) or a different steering layer? A sensitivity analysis would be crucial for understanding the robustness of your approach.

2. The performance recovery for Llama-3.1-8B on MATH500 is strikingly low (2.8%). You suggest this is due to "less clean steering directions." Does this imply that smaller models genuinely lack certain reasoning primitives, making your "knows how" hypothesis scale-dependent, or are the skills simply harder to isolate and activate in them?

3. Your taxonomy's quality metrics are derived from an LLM-as-a-judge. Have you considered a small-scale human evaluation to validate these automated metrics and ensure the discovered clusters represent meaningful cognitive functions rather than stylistic artifacts?

---

### Official Review · Reviewer_2nyo · 2025-10-27

**Soundness:** 2
**Presentation:** 4
**Contribution:** 2
**Rating:** 4
**Confidence:** 3

**Summary:**

This paper explores interventions in base models to make them perform like reasoning models. To achieve this, the authors trained small SAEs targeting reasoning features and steered the base models during generation using features from these SAEs, based on their activations in the reasoning model.

**Strengths:**

- The paper is well structured and easy to follow. All experiments are intuitive.
- The results are interesting for the community

**Weaknesses:**

- I am not sure how novel these results are. There are papers that have trained steering vectors as well [1], so for me, it was somewhat intuitive that base models are capable of performing reasoning with small interventions (thus, the knowledge of how to reason is already present in these models). It is much more fascinating to understand the limits of the knowledge in base models and whether RL could add more knowledge.

[1] https://arxiv.org/abs/2505.18706

**Questions:**

N/A

---

### Official Review · Reviewer_49fN · 2025-10-29

**Soundness:** 3
**Presentation:** 3
**Contribution:** 1
**Rating:** 2
**Confidence:** 4

**Summary:**

The paper argues that base language models already possess latent reasoning abilities and that reinforcement learning mainly teaches when to use them. Using SAEs to extract reasoning-related features, the authors derive steering vectors that, when applied appropriately, allow base models to recover most of the reasoning performance of thinking models without fine-tuning.

**Strengths:**

* The experiments span multiple architectures (Llama, Qwen, DeepSeek, QwQ) and benchmarks (GSM8K, MATH500), with ablation studies that isolate the effects of steering vector specificity
* Achieving up to 91% recovery of reasoning performance without any fine-tuning is both surprising and impressive

**Weaknesses:**

* The paper’s novelty is limited. From this, one could infer that the ability to reason already resides within the base model itself, and the proposed method primarily exposes or controls it rather than creating new capabilities. The latent steering approach closely parallels [1, 2], which similarly extracts reasoning-related direction vectors and reapplies them at inference to modulate reasoning depth. Thus, the methodological contribution is more incremental than groundbreaking, focusing on a conceptual reinterpretation rather than a new technical mechanism.

* The paper lacks a deeper analysis of its internal mechanisms — for instance, a more detailed examination of the steering vectors derived via SAEs would strengthen the interpretability and causal claims [3]

* The evaluation focuses primarily on math reasoning. It remains unclear whether similar latent reasoning applies to other domains

* The paper does not provide evidence of statistical significance for its reported results, leaving it unclear whether the observed improvements are robust or within the range of random variation.

[1] Liu et al, Fractional Reasoning via Latent Steering Vectors Improves Inference Time Compute

[2] Ward et al, Reasoning-Finetuning Repurposes Latent Representations in Base Models

[3] Mayne et al, Can sparse autoencoders be used to decompose and interpret steering vectors?

**Questions:**

See weaknesses

---

### Official Review · Reviewer_dWwX · 2025-10-30

**Soundness:** 2
**Presentation:** 3
**Contribution:** 2
**Rating:** 2
**Confidence:** 4

**Summary:**

The authors examine what reasoning training adds to a base model. They argue that the base model already possesses the necessary reasoning behaviours, and that reasoning training teaches it to activate them in the right places. They first cluster the model’s generated sentences with a small SAE and label each cluster with an LLM. They then train cluster-specific steering vectors (plus a shared bias) to maximize each sentence’s probability under the reasoning model. Finally, they add the appropriate steering vectors at sentences identified by the clustering mechanism and show that this improves the base model’s performance, closing up to 91% of the gap -- suggesting the model uses this mechanism after reasoning training.

**Strengths:**

* The paper is generally well written and easy to follow.
* The authors propose an interesting hypothesis that contributes to the ongoing discussion in the field.

**Weaknesses:**

### Some claims are unsupported
* **Pre-trained skills**

Trainable steering vectors do not necessarily correspond to “pre-trained skills.” Prior work such as ReFT [1] shows that even rank-1 representation edits can memorize hundreds of arbitrary input–output mappings. The fact that a set of such vectors, applied at appropriate places, achieves high accuracy does not imply that nothing new was learned.

* **Is the clustering necessary?**

The authors claim that the key addition of reasoning training is indicating when to activate reasoning mechanisms. However, [2] shows that training a single unconditional vector can match the performance of full-weight reasoning training. The “Only-Bias” ablation does not fully address this: both approaches train unconditional biases, differing mainly in how they are obtained. While the ablation supports clustering within the authors’ setup, it does not establish its universal necessity given the evidence in [2].

* **Moderate gains**
Beyond QwQ-32B (91% gain) and DeepSeek-R1-Distill-Qwen-32B (266% in a saturated regime), the hybrid approach yields limited improvements on other models. Even if the cluster-steering mechanism explains part of the reasoning performance, it appears far from explaining all of it on these models.

### Narrow Evaluation
* **Benchmark choice**

GSM8K and MATH500 are known to overlap with the pretraining data of, e.g., Qwen2.5 models, which already achieve high accuracy on these tasks with appropriate prompting prior to finetuning [3]. Consider evaluating on recent benchmarks such as AIME24/25 or AMC23.

* **Model coverage**

Qwen2.5 models are distinctive in achieving high performance after reasoning training and may not be representative of other families [4]. The only non-Qwen model evaluated is LLaMA, which shows a 2.8% gain (Table 2) with the proposed technique. Given the current experimental coverage, I remain skeptical about the generalizability of the claim.

* **Ablations**

Please run the ablation studies on all models and benchmarks used in the main experiments.

* **Labeling prompt bias**

I am concerned about Prompt 1 for labeling clusters, as it explicitly instructs labeling by reasoning topics, introducing bias. For example, why are there no “math” or “chemistry” clusters, which an SAE might discover? Using LLMs for labeling is acceptable, but as long as an LLM defines the labels, the approach remains unprincipled in the sense stated in the abstract.

---
### References
[1] Wu, Zhengxuan, et al. "Reft: Representation finetuning for language models." Advances in Neural Information Processing Systems 37 (2024): 63908-63962.

[2] Sinii, Viacheslav, et al. "Steering LLM Reasoning Through Bias-Only Adaptation." arXiv preprint arXiv:2505.18706 (2025).

[3] Liu, Zichen, et al. "Understanding r1-zero-like training: A critical perspective." arXiv preprint arXiv:2503.20783 (2025).

[4] Shao, Rulin, et al. "Spurious rewards: Rethinking training signals in rlvr." arXiv preprint arXiv:2506.10947 (2025).

**Questions:**

* If the SAE assigns multiple categories to a sentence, do you apply multiple steering vectors? If so, how are they combined?
* Please explain how you identify the elbow, and include illustrative figures in the appendix (method, criteria, and any thresholds used).
* Will the hybrid approach yield substantially worse gains when applied to low-scoring k–layer pairs (Figure 2)? If so, please quantify and explain why.

---

### Note · Authors · 2025-12-11

I have read and agree with the venue's withdrawal policy on behalf of myself and my co-authors.